# Supporting antidepressant discontinuation: the development and optimisation of a digital intervention for patients in UK primary care using a theory, evidence and person-based approach

Hannah M Bowers ![ORCID],[1] Tony Kendrick,[1] Marta Glowacka,[2] Samantha Williams,[1] Geraldine Leydon,[1] Carl May,[3] Chris Dowrick,[4] Joanna Moncrieff,[5] Rebecca Laine,[1] Yvonne Nestoriuc,[6] Gerhard Andersson,[7,8] Adam W A Geraghty ![ORCID] [1]

For numbered affiliations see end of article.

**Correspondence to**
Dr Hannah M Bowers;
h.m.bowers@soton.ac.uk

## ABSTRACT

**Objectives** We aimed to develop a digital intervention to support antidepressant discontinuation in UK primary care that is scalable, accessible, safe and feasible. In this paper, we describe the development using a theory, evidence and person-based approach.

**Design** Intervention development using a theory, evidence and person-based approach.

**Setting** Primary Care in the South of England.

**Participants** Fifteen participants with a range of antidepressant experience took part in 'think aloud' interviews for intervention optimisation.

**Intervention** Our digital intervention prototype (called 'ADvisor') was developed on the basis of a planning phase consisting of qualitative and quantitative reviews, an in-depth qualitative study, the development of guiding principles and a theory-based behavioural analysis. Our optimisation phase consisted of 'think aloud' interviews where the intervention was iteratively refined.

**Results** The qualitative systematic review and in-depth qualitative study highlighted the centrality of fear of depression relapse as a key barrier to discontinuation. The quantitative systematic review showed that psychologically informed approaches such as cognitive–behavioural therapy were associated with greater rates of discontinuation than simple advice to reduce. Following a behavioural diagnosis based on the behaviour change wheel, social cognitive theory provided a theoretical basis for the intervention. The intervention was optimised on the basis of think aloud interviews, where participants suggested they like the flexibility of the system and found it reassuring. Changes were made to the tone of the material and the structure was adjusted based on this qualitative feedback.

**Conclusions** 'ADvisor' is a theory, evidence and person-based digital intervention designed to support antidepressant discontinuation. The intervention was perceived as helpful and reassuring in optimisation interviews. Trials are now needed to determine the feasibility, clinical and cost-effectiveness of this approach.

## Strengths and limitations of this study

► A systematic review and qualitative meta-synthesis were conducted alongside primary qualitative work to guide the content of the intervention.

► A theory-based behavioural analysis and the development of guiding principles further informed the planning phase of intervention development.

► Think aloud interviews provided in-depth understanding of patients' views of the intervention in terms of usability and content.

► The intervention was iteratively refined throughout the think aloud interviews to produce an intervention that aligns with patient preference.

► Think aloud participants were predominantly white British and from more affluent regions in the South of England and may not represent the views of all antidepressant users.

## INTRODUCTION

The number of antidepressant prescriptions in the UK has continued to rise over the past four decades,[1] a trend which has also been seen in the USA and across Europe.[2,3] Approximately 10% of adults in the UK are currently prescribed antidepressant medication.[4] Though antidepressants can prevent relapse, there is evidence that 30%–50% of patients on long-term antidepressants have no indication based on guidelines for long-term use.[5–7] Research suggests this increase in prescribing is primarily due to general practitioners (GPs) prescribing antidepressants for longer and longer durations over time.[8] Long-term antidepressant use is both costly to the UK National Health Service (NHS) (in terms of prescription and appointment costs)

and is associated with increased side effects.[9] Attempting to discontinue antidepressants in the 30%–50% with no indication for long-term use may, therefore, be beneficial to patients and positively impact on use of healthcare resources.

There are many factors that may contribute to long-term antidepressant use, including the occurrence of a physiological withdrawal syndrome following reduction or cessation and psychological factors such as beliefs about the necessity of long-term use and fear of relapse.[10] Infrequent reviews of patients taking antidepressants may also contribute to sustained use.[11] However, simply prompting for patient reviews has resulted in discontinuation rates of 6%–8%, not significantly differing from usual care.[12 13] This highlights the potential importance of psychologically informed interventions to support withdrawal.

Trials have shown that cognitive–behavioural therapy (CBT) and mindfulness-based cognitive therapy (MBCT) can effectively support discontinuation of antidepressants, with cessation rates ranging from 55% to 95%.[14–18] Although producing positive outcomes, these interventions involve intensive group/face-to-face courses, thus access and ability to scale up within resource-strapped health services may be severely limited. There is a need for accessible, scalable psychologically informed interventions that can effectively support individuals where discontinuation is appropriate.

In the UK, 89% of the general population in 2018 used the internet weekly, up from 55% in 2006.[19] Internet-based digital interventions supported with human contact have been shown to effectively reduce depression and anxiety.[20] Digital intervention may have potential to provide a scalable, accessible way of supporting appropriate antidepressant discontinuation. We aimed to develop such a supported digital intervention as part of the UK-based REviewing long term antiDepressant Use by Careful monitoring in Everyday practice (REDUCE) programme to develop and trial safe, feasible and effective ways to support patients withdrawing from antidepressants where appropriate.

In this paper, we describe the planning and optimisation of our patient-facing digital intervention to support discontinuation, named 'ADvisor'. This paper provides an overview of the different stages of development and how these together informed a digital intervention. Some of this work has implications beyond intervention development and further details are, therefore, published elsewhere. This paper is instead focused on the particular work involved in developing a digital intervention.

## METHODS
### Phase 1: intervention planning and development
There is a range of systematic protocols for intervention development that can be drawn on at the outset of a development project (eg, intervention mapping[21]). We chose to implement a theory, evidence and person-based approach (PBA).[22] This comprehensive strategy integrates the PBA[23 24] with more commonly used theory and evidenced-based methods. The PBA provides guidance for integrating systematic in-depth qualitative research into the development process. Drawing on the PBA ensures evidence and theory-based techniques are applied with a full understanding of the target users' perspectives and psychosocial context.[23] We will outline the components of our comprehensive approach including systematic reviewing, primary qualitative research, development of guiding principles, behavioural analysis and logic modelling.

### Systematic reviewing
Two systematic reviews were conducted: a quantitative review with meta-analysis, and a qualitative thematic synthesis, described in detail elsewhere.[10 25]

The qualitative review searched nine databases from inception to February 2017 and updated searches were carried out in July 2018. Citation searching, reference list checking and related article checking were also performed. The quantitative review involved searching eight databases from inception to March 2017. Citations and reference lists were searched for full papers that met the inclusion criteria. Both searches were developed by an experience librarian and systematic reviewer. Further details of the search strategies can be found in the full publications of these reviews.[10 25]

For intervention planning, from the quantitative review, we drew out interventions that had successfully supported discontinuation and considered their intervention components, seeking full manuals where possible. We aimed to determine which components could be best translated into a digital format. In the qualitative review, we identified barriers and facilitators to antidepressant discontinuation. Barriers and facilitators were tabulated and used to inform the 'guiding principles' section as well as content for the intervention.

### Primary qualitative research
Individual semistructured interviews were conducted by SW with primary care patients with varying experiences of antidepressants, and varying levels of motivation to stop, with the aim to explore experiences of antidepressant discontinuation. These interviews explored patients' views on barriers and facilitators to withdrawal, the role of healthcare professionals in supporting withdrawal attempts and elements of a proposed intervention to support withdrawal. Interviews were conducted at the patients' homes or their GP practices and were audio recorded and transcribed verbatim. Patients provided written consent to take part. Analysis was conducted, following thematic analytical principles suggested by Braun and Clarke,[26] and Joffe and Yardley.[27] Analysis was conducted by SW (a qualitative researcher). The coding manual and developed themes were discussed and agreed by the wider development group. Only the findings related to the development of the intervention are described in this paper. Further details of the methods

and the findings related to the broader aims of this piece of qualitative work will be published elsewhere.

### Development of guiding principles

Guiding principles are a fundamental part of the PBA.[23] They represent broad design objectives that guide the application/implementation of the core intervention strategies, aiming to increase engagement.[24] Guiding principles were developed based on the qualitative synthesis[10] and primary qualitative findings. Through this qualitative work, we aimed to identify key behavioural needs, challenges or issues the intervention needed to address.

### Behavioural analysis

Behavioural and implementation theory was drawn on as we triangulated between the qualitative and quantitative evidence, and the expert views of our team (including patient representatives, GPs, psychiatrists, psychologists, sociologists and health services researchers) to determine important intervention components. Using the behaviour change wheel (BCW) and COM-B model of behaviour (capability, opportunity, motivation-behaviour),[28] informed by our qualitative research, we conducted a 'behavioural diagnosis'.[29] In behavioural diagnosis, factors that are likely to affect the central target behaviour are considered in terms of capability, opportunity, and motivation.[28 29] Once we had proposed initial intervention content/components, these were mapped theoretically using the BCW, social cognitive theory (SCT)[30] and normalisation process theory (NPT).[31] As well as providing a mapped full description of the proposed intervention, this process ensured we did not miss areas of theory that may have improved the intervention.

### Phase 2: intervention optimisation
#### Design

Within the PBA, 'think aloud' qualitative studies are employed to optimise the prototype intervention. Think aloud studies are designed to elicit in-depth perspectives about the nature of the content, rather than solely focusing on functionality and usability.

### Participants

Participants were recruited from eight primary care practices in the South of England. Eligibility criteria were as follows: inclusion criteria: Taking antidepressants for more than 1 year for a first episode or 2 years for a subsequent episode; discontinued antidepressants, or were in the process of tapering. Exclusion criteria: Patient Health Questionnaire (PHQ)-9 scores greater than or equal to 10 (suggesting persisting symptoms of depression) and those who reported any suicide ideation; history of suicide attempts; ongoing social difficulties or recent life events likely to provoke relapse; more than three previous significant episodes of depression; comorbid psychosis, bipolar disorder, obsessive–compulsive disorder or substance use (or

history of these conditions) or currently receiving psychiatric treatment.

### Procedure

Eligible participants met with a researcher (HMB, SW or TK) either in their own home or at their primary care practice where they provided written consent to take part in a think-aloud interview. Interviews invited participants to engage with the prototype intervention using a study laptop and say what they were thinking, aloud in real time. The interviewer prompted participants when necessary (eg, asking patients 'How do you feel about the information on this page?'). Interviews ranged from 38 to 93 min in length and were audio recorded and transcribed verbatim. The interview ended when patients concluded they had looked at all the information they would like to see or if the interview length was approaching 90 min. The amount of intervention content the patient saw, therefore, depended on their own preferences and the time they took to look at the information. The interview schedule can be found in online supplementary appendix A. There were three primary iterations of interviews based on three key modified prototype interventions. Patients at the start of the study, therefore, saw different versions of the intervention to those who were recruited later rounds. This allowed the changes made as a result of patient feedback to continue to be tested. Interviews with patients continued until data saturation was reached, defined here as when comments about the intervention reflected that no further changes were necessary according to the PBA and when there were no new codes identified as part of the thematic analysis.

### Analysis

Transcribed interviews were analysed using two primary analytic methods. The first analytical method was a more rapid coding than thematic analysis, which involves using coding tables designed for the PBA, where positive and negative comments were tabulated. Core problematic issues likely to affect participant engagement or intervention effectiveness identified using this coding method were brought to the broader group, and amendments to the intervention agreed. Alongside this method, a more in-depth thematic analysis[26 27] was developed to capture patient views of the intervention and ideas about how they might engage with it, beyond comments on what might be amended. For this latter analysis, HMB independently coded the transcripts and discussed a preliminary coding frame with a second researcher (AWAG). Theme labelling and interpretation were discussed and agreed by the team. The thematic analysis is presented here. Therefore, while the initial analysis informed what changes were necessary, the thematic analysis explored what patients thought about the intervention in greater depth. These analyses were related in that some things that were

**Table 1** Guiding principles for the ADvisor intervention

| ADvisor guiding principles | |
| --- | --- |
| **Design objectives** | **Key (distinctive) design features** |
| To build confidence that discontinuing antidepressant medication is safe and achievable over the long term | ► Offer an evidence-based rationale for how withdrawal and replacement with psychological/behavioural alternatives will help.<br>► Provide withdrawal success stories and examples (modelling).<br>► Address concerns patients may have re withdrawal (side effects, symptoms) from their previous experiences—demonstrate empathy and acknowledge real barriers to change.<br>► Offer motivational support. |
| To be an accessible, motivating resource that supports patients in managing their withdrawal in a manner that aligns with their preferences | ► Foster autonomy through choice and a non-prescriptive approach, providing explanations for all suggestions.<br>► Offer a broad range of strategies from quick support in managing withdrawal symptoms, to more in-depth modules on a mindful approach to preventing depression relapse, and behavioural strategies for managing day-to-day stressors.<br>► Provide options for self-tailoring to personal experiences and barriers<br>► Provide a simple, attractive interface, with a focus on accessibly of content |

identified in our initial analysis informed the development of themes.

## RESULTS

### Phase 1: intervention planning and development

#### Systematic reviewing

Our qualitative thematic synthesis (see ref. 10 for full results) across 22 studies highlighted key barriers and facilitators to discontinuation. Patients' concerns regarding their ability to cope and psychological dependence were common barriers, as were difficulties experienced in previous stopping attempts. Confidence in abilities to stop, effective coping strategies and stable life circumstances facilitated discontinuation. Additional important themes included fear of relapse—this was the central fear that prohibited stopping attempts—and beliefs about depression. The belief that depression was a long-term condition caused by biochemical changes in the brain was a key barrier to discontinuation. Where patients reported a very different belief that depression was due to changing life circumstances, this seemed to facilitate discontinuation. Patients' self-identity and goals were an important factor: Having self-identifying as 'old' or 'disabled' acted as a barrier to discontinuation, and having goals to function independently functioned as facilitator to discontinuation.

In the quantitative systematic review (see ref. 25 for full results), a variety of therapeutic techniques were implemented including a patient-specific letter to the GP with a recommendation to discontinue plus tapering advice; GP review of the patient's condition and medication; CBT plus tapering; MBCT with tapering support gradual discontinuation and 1-week tapering. The results indicated that CBT or MBCT plus tapering are helpful for patients discontinuing antidepressants, with cessation rates of 40%–95%,[23] compared with only 6%–8% cessation where health professionals are simply prompted to review patients. CBT plus tapering resulted in lower relapse rates compared with clinical management plus taper (15%–25% vs 35%–80%).[23] The content of the interventions was extracted and feasibility of delivery in a digital format was considered. We developed a module based closely on MBCT protocols on the basis of this review.

The findings from both reviews' findings informed the guiding principles, behavioural analysis and logic model, which formed the basis for intervention content selection and development.

#### Primary qualitative research

Five themes were developed through the thematic analysis of 19 patient interviews (full details will be published elsewhere). A summary is presented here. Participants spoke of the centrality of personal medication and healthcare factors, for example, some patients described the need for a personalised tapering regimen to support them discontinuing. Beliefs about depression and its treatment were key in shaping participants' stance towards discontinuing. For example, ideas around the necessity of antidepressant medication due to 'chemical imbalance' were common. Holding these beliefs made patients less likely to consider stopping. Fear of stopping, driven by fear of relapse were discussed as central barriers to withdrawal. The impact of others also appeared to be important. For example, the perception of stigma and the feeling of letting people down, made participants less willing to discontinue, while having a good support network was considered beneficial to stopping. Participants were also asked to consider digital methods of intervention delivery. Elements participants wanted to see in the intervention included explanation around how antidepressants work,

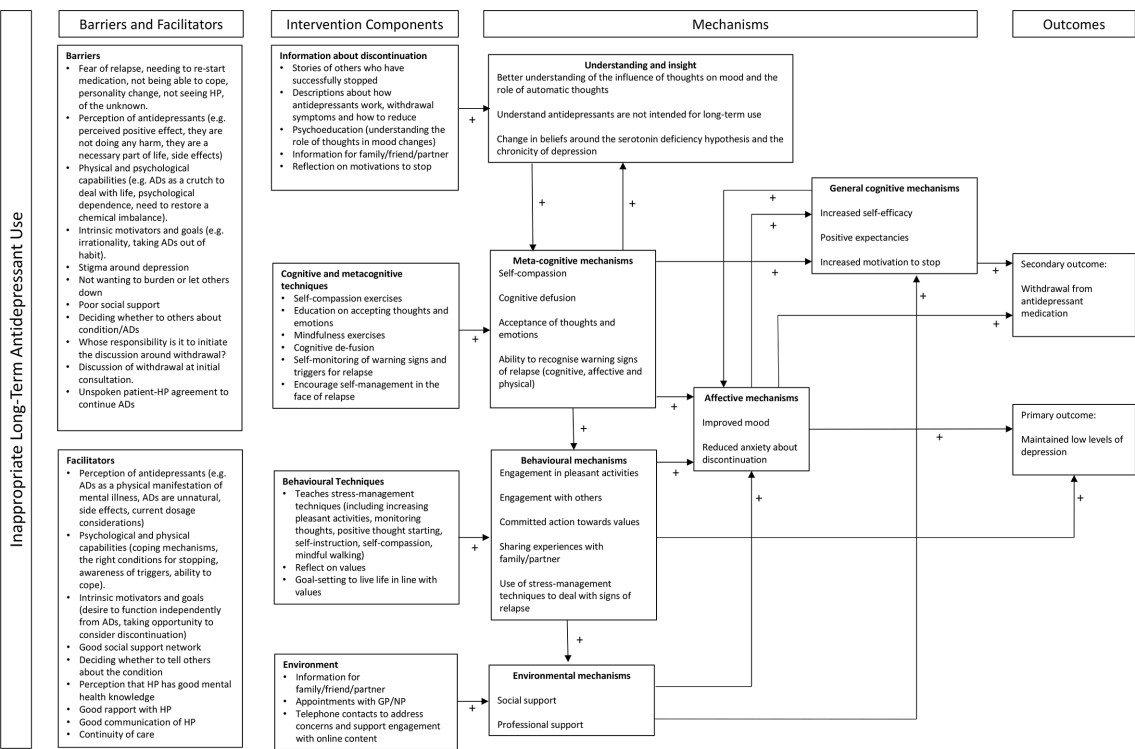

**Figure 1** Logic model advisor intervention alongside additional components. ADs, antidepressants; GP, general practitioner; HP, health professional.

support for anxiety/fear of discontinuing, coping strategies and information on withdrawal symptoms. There was some concern around privacy and around preference for greater face-to-face interaction to support them during the discontinuation phase. Patients expressed a need to have accessible, interactive and information presented in an aesthetically pleasing way.

The full findings in our primary qualitative research mirrored and expanded the findings of our qualitative thematic synthesis. They fed into the guiding principles, behavioural analysis, logic model and content for the intervention.

### Guiding principles

On the basis of the qualitative work, guiding principles were developed (comprised of design objectives and design features) (see table 1). We developed two broad design objectives: The first, regarding building confidence that discontinuing antidepressant medication is safe and achievable, was developed from prominent themes around fear of stopping, the need for confidence and beliefs that antidepressant medications are needed long term. The second objective that the intervention should be an accessible, motivating resource that supports patients in managing their withdrawal in a manner that aligns with their preferences, was developed in response to the range of views and beliefs held about the nature of depression and why antidepressants were necessary. Design features that support both these objectives are listed in table 1.

### Behavioural analysis

Our behavioural diagnosis following the COM-B model can be found in online supplementary appendix B. Our target behaviour was reducing and stopping the taking of antidepressant medication. Based on our reviews, qualitative work and discussion among our broader team, psychological capability and reflective motivation were considered key constructs for changing the target behaviour. The results of our behavioural diagnosis are presented in online supplementary appendix B.

Following the drafting of module content and structure, we mapped content against (1) studies suggesting content would be important, (2) BCW constructs, (3) SCT and (4) NPT. See online supplementary appendix C for detailed theoretical mapping for our intervention content.

Fundamentally, SCT[32] underlies the approach taken in the intervention to facilitate behavioural change. We ensured content aligned with the principles of SCT on how best to increase patient's confidence that they will be able to safely stop antidepressants (eg, drawing on persuasion, modelling and supporting performance exposure). We also focused on modifying outcome expectations, for example, increase positive expectation that the recommended strategies are likely to support effective discontinuation. At a later stage in development, the Necessity Concerns Framework (NCF)[33] was considered. NCF was developed to explain the role of treatment beliefs on adherence behaviours. According to NCF, adherence to treatment is a function of patients' beliefs about the

**Table 2**  Outline content of the digital intervention

| Content | Description |
| --- | --- |
| Reducing and stopping antidepressants | An introduction to the intervention, which addresses motivations behind withdrawal, asking participants to reflect on why they might prefer to discontinue antidepressant treatment. Guidance on when to speak to their GP/nurse and advice on following a tapering regimen. |
| Thinking about antidepressants | Acknowledging that antidepressant treatment is not necessarily required long term and that the mechanisms are more complex than correcting a serotonin deficiency. |
| I'm worried about stopping | Addressing participant fears by signposting participants to appropriate resources in ADvisor. |
| Dealing with withdrawal symptoms | Guidance for dealing with mild withdrawal symptoms (including guided practices for accepting/tolerating unpleasant symptoms). Advice for patients to contact their GP for assistance with moderate or severe withdrawal symptoms. |
| Keeping well | Relapse prevention techniques grounded in mindfulness-based cognitive therapy. |
| Thinking about what you value | Reflection on values and committed action to values (through goal setting), based on acceptance and commitment therapy. |
| Moving forward | Psychoeducation and techniques for managing distress (eg, mindfulness and behaviour activation) provided through a distress-management online intervention, Healthy Paths. |
| My notes | Where patients can access content from other sections where they have written their own responses (eg, their own reasons for wanting to stop antidepressants and their own warning signs and triggers for relapse). |
| Resources | Direct links to resources in ADvisor (eg, activity planning and information for family and friends). |

GP, general practitioner.

necessity of their medication and the concerns they have about it; high necessity beliefs and low concerns are likely to predict medication adherence.[34] In the context of antidepressant withdrawal, accordingly, we would need to reduce patients' beliefs about the necessity of the medication, highlight likely benefits of stopping and reduce concern regarding the stopping process. All of these factors will ultimately impact on self-efficacy, hence the centrality of SCT in our theoretical modelling.

### Logic modelling

Logic models represent proposed or hypothesised 'theories of change' outlining the problem/issue and barriers, ingredients mechanism and how these may affect target outcomes.[35] We developed a draft logic model for the REDUCE patient intervention, drawing on theory, evidence-based and our person-based qualitative work, see figure 1.

### Outline intervention content

On the basis of our planning process, a prototype digital intervention was developed for patients taking antidepressants long term (defined as more than 1 year for a first episode or more than 2 years following two or more episodes). The contents of the online intervention are described in table 2. A digital intervention for health professionals (providing information and guidance on antidepressant reduction) was also developed as part of the REDUCE programme and is reported separately.

Content was developed using findings from the reviews of the literature, primary qualitative work, behavioural analysis and logic modelling. In addition to online content, scheduled telephone support contacts with specialists trained in providing psychological support and email reminders were developed as part of the patient intervention.

When accessing the ADvisor intervention for the first time, users view a core module with the central rationale for stopping antidepressants; they can then access a menu with a range of further modules based on our planning work. Aligning with our guiding principles, users are advised that they can use ADvisor how and when they would like. It is their tool, to be used to support them in a way that is consistent with their needs, preferences and experience. Through this approach we aimed to maximise autonomous motivation.[36]

Content for the online intervention was initially drafted by a member of the content development team (HMB) before AWAG and MG and then wider team members offered their expertise and informed further development of the content. This iterative process continued until all team members were satisfied that the prototype intervention addressed key experiences, barriers and facilitators identified by the work from phase one and were in line with the guiding principles, theoretical modelling and logic model. The content was transferred into online pages in LifeGuide (www.lifeguideonline.org) and further amendments to the presentation were made by the team before moving forward to the optimisation phase.

### Phase 2: intervention optimisation

Of the 42 patients who returned a postal reply slip expressing interest, 11 were ineligible, 9 could not subsequently be contacted, 2 later declined and 5 expressed an

Table 3 Think aloud qualitative study characteristics

| Characteristics | N (%) |
|---|---|
| Females | 9 (60) |
| Males | 6 (40) |
| Married | 11 (73.3) |
| cohabiting | 2 (13.3) |
| Single | 2 (13.3) |
| Employed | 9 (60) |
| Not currently in employment | 6 (40) |
| Diagnosis | |
| Depression/low mood | 9 (60) |
| Fibromyalgia | 2 (13.3) |
| Unknown | 2 (13.3) |
| Urethritis | 1 (6.7) |
| Post-traumatic stress disorder | 1 (6.7) |
| Successfully stopped before | 8 (53) |
| Currently taking antidepressants | 14 (93.3) |
| | **Mean (SD)** |
| Age | 55.20 (15.59) |
| Years on antidepressants | 10.43 (7.27) |
| PHQ-9 score | 4.53 (2.50) |

PHQ-9, Patient Health Questionnaire - 9.

interest only after data saturation had been reached. This resulted in a final sample of 15 patients (see table 3 for sample characteristics).

## Iterations of advisor

There were three rounds of iterations of the intervention during the think-aloud interviews. Patients in round 1 were shown the first prototype. Changes made to the version in round 2 included making the tone less formal, revising the introduction navigation and the wording to be more gentle. The 'my notes' section was also reorganised to be clearer and buttons to exit the intervention at the end of each module were removed to try to keep the patients on site for longer. In the version shown in round 3, some changes included further revision of the tone, some of the information was presented in a more aesthetically pleasing way and some links within the intervention to other modules were removed as these were confusing for patients.

## Findings

Six themes were developed, namely: flexible use; familiarity with content; reassurance; utility of information; teaching of useful skills and feeling supported. Patient identifiers and demographic information are presented below each quote, where round number refers to the iteration of the intervention that the patient saw.

## Flexible use

Participants discussed how ADvisor could be used in different ways to suit the individual. When viewing the main menu page in ADvisor participants talked about how different sections would be more useful for them, and that some sections were not relevant for them at that particular time.

> Dealing with withdrawal symptoms, I don't have any, so it's fine. That [keeping well and moving forward modules] I'm more interested in about because I think that's—for me, keeping well and moving forward is where I am and where I want to be.

> [14/03/0001] [round 1]

Initial versions of the intervention included an introduction module within which participants could choose which of two options they would like to view first, though they would need to view both sections before moving onto the main menu. Some participants felt that this was in contradiction to the aim of choice and flexibility. We, therefore, modified the intervention so that the introduction was shorter and these two choices were moved to optional buttons in the main menu.

> It's kind of saying you've really got to look at that one; otherwise, you will have flicked back through or I would have thought it might have been, if it's really flexible, user friendly, you might be allowed to skip that page because you could always revisit it again.

> [01/01/0026] [round 1]

Participants not only varied in the topics they wanted to look at, but also in terms of the different exercises they would choose to engage with in ADvisor. Some participants liked the idea of writing down their responses in ADvisor while others did not.

> No. That's me. No, I'm very stoic and—just—I don't need to write it down, it's fine; I know what I'm doing, I'm fine, very much, I think.

> [01/01/0005] [round 2]

> I'd like to say that I would [write things down]; I think I probably would if I was—you know—really serious about it, because I like to write things down and if I haven't written it down, it can just go out of my brain. So I think, for me, it would be important to write that down.

> [05/01/0022] [round 2]

Participants also discussed how ADvisor could be used in different ways. For example, it can be something used regularly, something one can pick up as and when necessary or it can be read through all in one go.

> So it looks like you can use it when you want to but if you feel you're coping without, so it's not something you have to do all the time.

> [05/01/0022] [round 2]

Yes, I would use it for future reference, as well, because you can always go backwards, can't you? With anything, I mean. If I ever came to a time where I was feeling down, I think, to go back on to something is to remind you. Because it's easy to forget.

[13/01/0058] [round 3]

### Familiarity with content

Many of the participants referred to previous experience with psychological therapies or tools they have used in the past for their symptoms of depression. When reading cognitive-behavioural, acceptance and commitment, or mindfulness-based information in ADvisor, participants expressed a sense of familiarity with the terminology or messages they were presented.

Clicking on Breathing Space; that's very much mindfulness, isn't it? Yes, I like that, that's nice.

[14/03/0001] [ round 1]

Some of the information about depression and antidepressants seemed to be obvious to a small number of participants who had pre-existing knowledge, but they understood that not all patients would have the same prior knowledge. One participant in particular who worked in healthcare found that much of the information was not new to her.

I'm obviously interested in reducing still further or coming off the antidepressants. … See I don't think I can—I do know an awful lot about it and read a lot about it and very—sorry—but, you know, being in the business myself, it's all a bit Noddy to Big Ears.

[13/01/0033] [round 3] [works in healthcare]

### Reassurance

Participants described a sense of fear around stopping antidepressants. This has been reported in previous qualitative studies of patient and health professional perspectives on stopping.[10] Participants in this study often reported feeling reassured by information in ADvisor. While participants differed in terms of which particular piece of information they found reassuring, some participants noted feeling reassured knowing that they could go back on their antidepressant if they felt necessary. Other participants found that knowing that withdrawal symptoms are often short-lived offered reassurance.

Well that's a good section because that is quite a worry, I think, for anybody wanting to come off them; it would worry me what would my side-effects be and how would I feel coming off them. So to actually—I mean I didn't know this—to actually say that they are often short lived and go away in a few days or weeks is quite encouraging, isn't it.

[04/01/0025] [round 3]

As fear of withdrawal symptoms was highlighted in the qualitative work, withdrawal symptoms were discussed at several points during the introduction module. However,

participants who were not initially concerned about withdrawal symptoms felt that this was setting an expectation for difficulty withdrawing. While not minimising withdrawal-related problems, we, therefore, revised the language around concerns about withdrawal in the introduction.

Well it's very obvious withdrawal is a problem, looking at all the advice you can see to help you get over it, which—yes. There's a negative feeling there, if it's stressed to this degree on this program, then you're obviously expecting trouble.

[10/03/0003]

Credibility of the information appeared to be important for participants. Participants liked to see the evidence base that was provided in ADvisor and in particular liked that it would be used within an NHS setting. The NHS affiliation seemed to provide a sense of reliability and credibility.

I'd be really pleased if they [GP/nurse] referred me to a website, especially if it was from the GP, because I think, well, it's backed up or supported by them.

[14/03/0001] [round 1]

There was a balance that needed to be struck between portraying information as credible and maintaining a warm and friendly tone. Participants reported some of the information in ADvisor as sounding academic and reading like it could be used by practitioners. As a result, the tone was revised to be warmer and friendlier, while maintaining a sense of credibility.

It's just very business-like so very much like maybe something that a university would produce or maybe that a medical professional would share amongst themselves and your everyday person who's maybe not used to reading things in so much detail any more, sadly. It's quite dry.

[14/03/0001] [round 1]

### Utility of information

Participants described the information on withdrawal symptoms to be useful, in particular, some participants liked the information on how to distinguish between signs of relapse and withdrawal symptoms. One participant in particular expressed a shift in her views on discontinuing as a result of the information in ADvisor. She explained that had she known that withdrawal symptoms may feel like relapse and will pass, she may have persisted with her lower dose of antidepressant for longer. She also highlighted that difficulty in getting a GP appointment is a barrier for her to persist with discontinuing in the face of difficulties.

… I didn't know … withdrawal symptoms might appear the same as the symptoms that led to needing antidepressants in the first place, but they will pass after a short time; I didn't know that. I thought if you

started feeling down again, then you were heading for a crash.

[13/03/0001] [round 2]

Some participants described wanting more detailed information about what withdrawal symptoms might be expected. However, on discussion with the broader study team, it was decided to avoid setting expectations around particular symptoms as this may lead patients to experience expected symptoms. Patients can instead request this information from their GP if it is something they feel they would rather know about. While this information is provided to GPs as part of our health professional intervention package, it must be acknowledged that there are limitations around access to GP appointments, which may act as a barrier to getting information about withdrawal symptoms.

Participants also noted that it was useful to reflect on the side effects of taking antidepressants. There was an awareness that these can be hard to recognise, and three participants reported that after reading the information in ADvisor, they may in fact have been experiencing side effects of which they were previously unaware. One participant described how this made him even more inclined to discontinue.

Well, as I look at these, I think maybe I'm wrong; maybe I am still getting side-effects, but I've just learned to accept them or—I'm just a little bit in denial and it makes me want to get off them even more, because then—lots of these things will, you know, will disappear.

[12/03/0003] [round 1]

### Teaching of useful skills

Participants reported the skills included in ADvisor as being useful. In particular, advice around preventing relapse and mindfulness-based skills were considered to be useful.

Your triggers, recognising your emotions and reminding yourself that you don't have to react in a certain way; you can react in a different way. Yes, I think it's very good.

[13/01/0001] [round 2]

Acceptance of difficulties and of emotions was discussed as a useful coping strategy by participants, both with regards to their own pre-existing relationship to their emotions, and with regards to the messages in ADvisor on acceptance.

When you read it like that, it is true; the more you worry about things, the more down you get. So you've got to learn to stop doing that. I have to start putting that into practice if I'm going to do this.

[13/01/0058] [round 3]

Participants liked having tools and techniques in ADvisor for dealing with difficult emotions and life

stresses. There was an understanding that life stress is often unavoidable, and participants expressed a desire to learn ways of dealing with stresses. Some participants stated that learning how to manage emotions would act as a replacement for taking antidepressants.

I think that exercise of sitting by the stream is very good, because I know when I had Cognitive Behavioural Therapy I was taught to—you know—when your thoughts came—to—and I still do this now—is always remember—say to yourself that it will pass, those feelings will pass and it might be horrible while you're going through those feelings, but find somewhere nice and comfortable to sit, with a blanket even, and that sort of thing.

[04/01/0025] [round 3]

By the final interviews in the final round, participants' comments were positive with no new issues being identified. This signified the intervention was now ready for further evaluation and feedback in the planned feasibility trial to follow.

## DISCUSSION

We developed a digital intervention to support appropriate antidepressant discontinuation. The intervention was developed through a process of triangulation between quantitative and qualitative review evidence, theory and in-depth qualitative research. 'ADvisor' is designed to support ways of understanding antidepressants and to help people to withdraw more successfully. It provides resources to build confidence for, and to support, stopping including side effect management, addressing concerns, depression relapse prevention and stress management. The application of the PBA[22–24] has ensured our intervention is grounded a rich understanding of patients' psychosocial context.

Discontinuation can be complex,[10] and the digital ADvisor intervention is designed to be an information-based resource to support patients, alongside monitoring and review from their (GP, family doctor). A separate digital intervention has been developed for GPs and other primary care professionals, called 'ADvisor: Health Professionals'. The patient intervention will also be used with additional brief telephone guidance (up to an hour, spread over three calls by trained psychological practitioners), to support use of the material. Guided digital/internet-based resources have been found to be consistently more effective than unguided digital interventions[37] for mental health problems. Guidance in this context is especially important as patients are withdrawing from pharmacotherapy, thus close monitoring is necessary.

The intervention will be implemented in a feasibility randomised controlled trial, where we will carry out a full qualitative[38] and quantitative[35] process study. We will explore how people engage with the intervention and how it affects their discontinuation experience. On this

basis, as in the latter stages of the PBA,[24] we will continue to modify the intervention ahead of a fully powered main trial.

There are some limitations to consider. Our recruitment for our qualitative work was from a limited, relatively affluent, geographical area in the south of England. The majority of our participants were women in both the primary qualitative work and the think-aloud interviews. While this does reflect the higher rates of antidepressant use for depression in women,[39] it may be that our findings do not accurately reflect the views of men on long-term antidepressants. In the think-aloud interview sample, only 9 of the 15 participants were taking antidepressants long term for depression or low mood. The intervention contains information on preventing depression relapse and focuses on the symptoms of depression and anxiety which may not be applicable to these individuals. As such, some members of our sample may not have adequately represented the target population for this intervention, which may have introduced bias in our findings. The average age of participants in our think-aloud interview sample was 55.2 years, which may be a reflection of the typical populations in the geographical locations in this study. In the feasibility trial and main trial phases of intervention testing, further qualitative work will be carried out with a larger and demographically wider population of patients from a range of different areas in the UK.

The researchers conducting the think-aloud interviews were involved in the development of the intervention. This may have resulted in bias when asking questions about the intervention. However, in think-aloud interviews, the patients often express their views in response to what they see on the page as opposed to solely responding to questions from the researcher. While prompting and follow-up questions might have been affected by researcher bias, patients were not aware the interviewers had designed and written elements of the intervention and were encouraged to provide both positive and negative feedback to the researchers.

To conclude, psychologically informed interventions may improve the chances of effective discontinuation from antidepressants. ADvisor is a theory, evidence and person-based digital intervention that may provide this support. The feasibility, clinical and cost-effectiveness of ADvisor now needs to be determined.

**Author affiliations**
[1]Primary Care, Population Sciences and Medical Education, University of Southampton, Southampton, UK
[2]Faculty of Health and Social Sciences, Bournemouth University, Poole, UK
[3]Faculty of Epidemiology and Population Health, London School of Hygiene and Tropical Medicine, London, UK
[4]Institute of Population Health Sciences, University of Liverpool, Liverpool, UK
[5]Mental Health Sciences, University College London and North East London mental health trust, London, UK
[6]Department of Psychology, Helmet Schmidt University, Hamburg, Germany
[7]Department of Behavioural Sciences and Learning, Linkoping University, Linköping, Sweden
[8]Department of Clinical Neuroscience and Psychiatry, Karolinska Institutet, Stockholm, Sweden

**Acknowledgements** The ADvisor was developed using LifeGuide software and / or methodologies, which was partly funded by the NIHR Southampton Biomedical Research Centre (BRC). The authors would like to acknowledge the work of Emma Maund while working on the REDUCE Programme, who conducted two systematic reviews which informed the intervention development.

**Contributors** TK led on the grant application for the 6-year REDUCE programme. SW conducted primary qualitative interviews which informed the intervention content. AWAG led the development of the intervention. AWAG and HMB conducted theoretical modelling, behavioural analysis and developed guiding principles. HMB drafted intervention content and discussed with the intervention development team (AWAG and MG) and the wider team (TK, SW, GL, CM, CFD, JM, RL, YN and GA). MG developed the intervention into a digital format using Lifeguide software and led on intervention testing. Think aloud interviews were conducted by HMB, SW and TK. RL provided support with recruitment for think aloud interviews. Think aloud transcripts were coded by HMB and the results were discussed with AWAG, GL, TK and CM for interpretation. HMB, MG and AWAG refined the intervention in line with patient feedback, with comments from the wider team when necessary. The manuscript was prepared by HMB and AWAG, and has been approved by all coauthors.

**Funding** This work was supported by NIHR Programme Grant for Applied Research (PGfAR) grant number RP-PG-1214-20004.

**Competing interests** TK reports grants from National Institute for Health Research, during the conduct of the study. JM reports grants from National Institute of Health Research, during the conduct of the study; and is a member of the Council for Evidence-based Psychiatry which is an unfunded organisation, whose mission is to 'communicate evidence of the potentially harmful effects of psychiatric drugs to the people and institutions in the UK that can make a difference'.

**Patient consent for publication** Not required.

**Provenance and peer review** Not commissioned; externally peer reviewed.

**Data availability statement** No data are available. This is a qualitative study and therefore the data are not suitable for sharing beyond what is contained within the report. Further information can be requested from the corresponding author.

**ORCID iDs**
Hannah M Bowers http://orcid.org/0000-0002-1996-6652
Adam W A Geraghty http://orcid.org/0000-0001-7984-8351

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
