## [Reviewer comments · BMJ Open]

ARTICLE DETAILS

TITLE (PROVISIONAL)	Supporting antidepressant discontinuation: The development and optimisation of a digital intervention for patients in UK primary care using a theory-, evidence-, and person-based approach
AUTHORS	Bowers, Hannah M; Kendrick, Tony; Glowacka, Marta; Williams, Samantha J; Leydon, Geraldine; May, Carl R; Dowrick, CF; Moncrieff, Joanna; Laine, Rebecca; Nestoriuc, Yvonne; Andersson, Gerhard; Geraghty, Adam W A

VERSION 1 – REVIEW

REVIEWER	Xiancang Ma Xi'an Jiaotong University
REVIEW RETURNED	30-Aug-2019

GENERAL COMMENTS	The manuscript entitled “Supporting antidepressant discontinuation: The development and optimization of a digital intervention for patients” showed a digital intervention to support antidepressant discontinuation in UK primary care. This paper is a meaningful and well-written paper, There are some comments below: In this study, only 15 participants was selected after exclusion criteria. Besides, the participants in this study was mainly in white British. I propose to expand the sample size in the follow-up study. More participants may provide more information for comparison and analysis.
--

REVIEWER	Marloes Huijbers Radboud University Medical Center, Nijmegen, the Netherlands
REVIEW RETURNED	01-Sep-2019

GENERAL COMMENTS	The current manuscript describes different stages in the development of a digital intervention to support people to come off their antidepressant medication (ADM), conducted in the UK. The clinical merit of this study can be considered large, as there seems to be a clear need for more knowledge and practical support regarding discontinuation ADM, both for patients and clinicians. Below I will outline my major comments that might serve to further improve the manuscript. 1. To me, the most important hindrance in reading the article was the scope of this project, with several elements all put into one manuscript. The breadth of information and different theoretical models applied were somewhat overwhelming to me as a reader, and sometimes confusing (“ what belongs where?”). More importantly, the full findings of the qualitative study are missing
---

	here, which apparently are key to all subsequent stages and models. a. I would consider either publishing the qualitative study first and referring to it (saving space and information overload in the methods and results sections) before moving on to the intervention optimisation phase, or consider including the full qualitative methods and results here, rather than elsewhere. In the latter case, the intervention optimisation phase might need to be described somewhat more concise, also in light of the authors' future intention to carry out a full qualitative and quantitative process study. b. The previously published information (qualitative and quantitative reviews) could be integrated into the introduction, so as not to take up space in the methods and results sections. c. Although I trust that every theoretical model has a valuable contribution to the intervention development, I wondered if there might be too many and too prominent here, in disadvantage of (and absence of) actual content. d. The information provided in Fig. 1 and Tables 1 and 2 (and appendices) seem to be the outcome of a "black box" because the more detailed descriptions of procedures etc are missing. 2. On page 27, the authors describe how the team decided to leave out detailed descriptions of withdrawal symptoms to avoid setting negative expectations, and instead referring patients to their GP for further guidance. However, on page 26 concerns were raised about the accessibility of GP care and how this could be a hindrance to proceed with discontinuation. The authors might reflect on this in the discussion, and possibly explore alternative options (are there others who might provide this care and who are more easily contacted, preferably the same day?). 3. The manuscript is generally written in a clear and fluent style, however at some points appears slightly colloquial. Some sentences could be improved to a more academic level.
--	--

REVIEWER	Dr Tom Kingstone School of Primary, Community and Social Care, Keele University, UK
REVIEW RETURNED	29-Sep-2019

GENERAL COMMENTS	Dear authors, A fantastic body of work and a very commendable effort to fit a lot of information and process into this manuscript. I read with much interest and would like to suggest the following amendments: Abstract: The statement of objectives in the abstract does not match those stated at the end of the intro/background section or the title of the paper. Please refer to optimisation for consistency. Methods: description of the 'think aloud' method seems a little light in detail. For example, how much of the ADvisor intervention was covered in each interview? Were participants given a laptop or did they use their own PCs (important for familiarity, and recruitment)? Recruitment: Do the authors mean >10 or >= 10 (suggesting moderate level of depression and/or depression not under control). Some details are missing in the sample characteristics: age, education level, level of internet use (if asked)?
---

	Analysis: I was a little confused with the description of the two processes of analysis in the qualitative study. How did they overlap and/or inform one another? In terms of saturation, the definition seems to apply more to the rapid coding approach than to the thematic approach. Findings: I would have found it helpful to have another flowchart to present the key changes made to the ADvisor intervention across the interviews; to describe the different iterations and, most importantly, the process of optimisation (unless this is reported elsewhere?). No reporting standards or guidelines for qualitative research are reported.
--	--

VERSION 1 – AUTHOR RESPONSE

Reviewer: 1

Reviewer Name

Xiancang Ma

Institution and Country

Xi'an Jiaotong University

Please state any competing interests or state 'None declared':
None declared

Please leave your comments for the authors below The manuscript entitled “Supporting antidepressant discontinuation: The development and optimization of a digital intervention for patients” showed a digital intervention to support antidepressant discontinuation in UK primary care. This paper is a meaningful and well-written paper, There are some comments below:
In this study, only 15 participants was selected after exclusion criteria. Besides, the participants in this study was mainly in white British. I propose to expand the sample size in the follow-up study. More participants may provide more information for comparison and analysis.

Thank you for highlighting this. We have added to the discussion that future research will explore patient views using a more diverse sample:

Page 20 line 18-32:

“There are some limitations to consider. Our recruitment for our qualitative work was from a limited, relatively affluent, geographical area in the south of England. The majority of our participants were women in both the primary qualitative work and the think-aloud interviews. While this does reflect the higher rates of antidepressant use for depression in women [39], it may be that our findings do not accurately reflect the views of men on long-term antidepressants. In the think-aloud interview sample, only nine of the 15 participants were taking antidepressants long-term for depression or low mood. The intervention contains information on preventing depression relapse and focuses on the symptoms of depression and anxiety which may not be applicable to these individuals. As such, some members of our sample may not have adequately represented the target population for this intervention, which may have introduced bias in our findings. The average age of participants in our think-aloud interview sample was 55.2 years, which may be a reflection of the typical populations in the geographical

locations in this study. In the feasibility trial and main trial phases of intervention testing, further qualitative work will be carried out with a larger and demographically wider population of patients from a range of different areas in the UK.”

Reviewer: 2

Reviewer Name

Marloes Huijbers

Institution and Country

Radboud University Medical Center, Nijmegen, the Netherlands

Please state any competing interests or state ‘None declared’:
None declared

Please leave your comments for the authors below The current manuscript describes different stages in the development of a digital intervention to support people to come off their antidepressant medication (ADM), conducted in the UK. The clinical merit of this study can be considered large, as there seems to be a clear need for more knowledge and practical support regarding discontinuation ADM, both for patients and clinicians.

Below I will outline my major comments that might serve to further improve the manuscript.

1. To me, the most important hindrance in reading the article was the scope of this project, with several elements all put into one manuscript. The breadth of information and different theoretical models applied were somewhat overwhelming to me as a reader, and sometimes confusing (“ what belongs where?”). More importantly, the full findings of the qualitative study are missing here, which apparently are key to all subsequent stages and models.

a. I would consider either publishing the qualitative study first and referring to it (saving space and information overload in the methods and results sections) before moving on to the intervention optimisation phase, or consider including the full qualitative methods and results here, rather than elsewhere. In the latter case, the intervention optimisation phase might need to be described somewhat more concise, also in light of the authors’ future intention to carry out a full qualitative and quantitative process study.

Thank you for this comment. We appreciate that this paper covers a lot of ground, including qualitative work that will be reported elsewhere. We have included the development-specific qualitative research in the methods and results of this paper. We are keen to keep this material and level of detail in this paper as we wish to transparently describe all aspects of our development approach. Additionally, the first and third reviewer were happy with this form of presentation.

Nonetheless, to improve the clarity around this issue, we have revised the discussion of the aims and focus of the paper in the introduction and we have clarified the aims of the qualitative interviews: We have explained that the interviews were carried out as a stand-alone study with their own broader aims, but that the findings were used to inform the intervention development. We hope that our amendments have provided further clarity around this issue.

Page 5 line 1-10:

“We aimed to develop such a supported digital intervention as part of the UK-based REDUCE (REviewing long term antiDepressant Use by Careful monitoring in Everyday practice) programme to develop and trial safe, feasible and effective ways to support patients withdrawing from antidepressants where appropriate.

In this paper we describe the planning and optimisation of our patient-facing digital intervention to support discontinuation, named 'ADvisor'. This paper provides an overview of the different stages of development and how these together informed a digital intervention. Some of this work has implications beyond intervention development and further details are therefore published elsewhere. This paper is instead focused on the particular work involved in developing a digital intervention."

Page 6 line 12-24:

"Individual semi-structured interviews were conducted by SW with primary care patients with varying experiences of antidepressants, and varying levels of motivation to stop, with the aim to explore experiences of antidepressant discontinuation. These interviews explored patients' views on barriers and facilitators to withdrawal, the role of health care professionals in supporting withdrawal attempts, and elements of a proposed intervention to support withdrawal. Interviews were conducted at the patients' homes or their GP practices and were audio recorded and transcribed verbatim. Analysis was conducted following thematic analytic principles suggested by Braun and Clarke [26], and Joffe and Yardley [27]. Analysis was conducted by SW (a qualitative researcher). The coding manual and developed themes were discussed and agreed by the wider development group. Only the findings related to the development of the intervention are described in this paper. Further details of the methods and the findings related to the broader aims of this piece of qualitative work will be published elsewhere."

b. The previously published information (qualitative and quantitative reviews) could be integrated into the introduction, so as not to take up space in the methods and results sections.

Thank you for this suggestion. We felt it was necessary to include the systematic reviews in the methods and results of this paper as these form an important part of the development work that the paper is describing. While these reviews have broader aims and implications that are reported elsewhere, they were a part of the methods of the work reported in this manuscript. We have reserved the introduction for discussion of prior research that was not part of the current intervention development methods. Furthermore, the editor's comments suggest that further information about the methods of the reviews was necessary.

To try to improve the understanding and clarity around this, we revised the aims and focus of the paper in the introduction

Page 5 line 1-10:

"We aimed to develop such a supported digital intervention as part of the UK-based REDUCE (REviewing long term antiDepressant Use by Careful monitoring in Everyday practice) programme to develop and trial safe, feasible and effective ways to support patients withdrawing from antidepressants where appropriate.

In this paper we describe the planning and optimisation of our patient-facing digital intervention to support discontinuation, named 'ADvisor'. This paper provides an overview of the different stages of development and how these together informed a digital intervention. Some of this work has implications beyond intervention development and further details are therefore published elsewhere. This paper is instead focused on the particular work involved in developing a digital intervention."

c. Although I trust that every theoretical model has a valuable contribution to the intervention development, I wondered if there might be too many and too prominent here, in disadvantage of (and absence of) actual content.

Thank you for bringing this to our attention. We have revised our results section on the theory-driven elements of intervention development so that it now provides an overview of how the theories are used and refers the reader to the appendices for detailed explanations of how the different components in each theory relate to the content of the intervention. While we mention all theories, we now primarily focus on SCT and why this was a core framework for our intervention, in line with this comment.

Page 11 line 10-33:

“Our behavioural diagnosis following the COM-B model can be found in Appendix A. Our target behaviour was reducing and stopping the taking of antidepressant medication. Based on our reviews, qualitative work and discussion amongst our broader team, psychological capability and reflective motivation were considered key constructs for changing the target behaviour. The results of our behavioural diagnosis are presented in Appendix A.”

“Following the drafting of module content and structure, we mapped content against 1) studies suggesting content would be important, 2) Behaviour Change Wheel (BCW) constructs, 3) Social Cognitive Theory (SCT), and 4) Normalisation Process Theory (NPT). See Appendix B for detailed theoretical mapping for our intervention content.”

“Fundamentally, SCT [32] underlies the approach taken in the intervention to facilitate behaviour change. The intervention is designed to increase self-efficacy for stopping and to modify outcome expectations e.g. increase positive expectation that the recommended strategies are likely to support effective discontinuation. At a later stage in development, the Necessity Concerns Framework (NCF) [33] was considered. NCF was developed to explain the role of treatment beliefs on adherence behaviours. According to NCF, adherence to treatment is a function of patients’ beliefs about the necessity of their medication and the concerns they have about it; high necessity beliefs and low concerns are likely to predict medication adherence [34]. In the context of antidepressant withdrawal, accordingly, we would need to reduce patients’ beliefs about the necessity of the medication, highlight likely benefits of stopping, and reduce concern regarding the stopping process. All of these factors will ultimately impact on self-efficacy, hence the centrality of SCT in our theoretical modelling.”

d. The information provided in Fig. 1 and Tables 1 and 2 (and appendices) seem to be the outcome of a “black box” because the more detailed descriptions of procedures etc are missing.

Unfortunately, we are not sure what exactly reviewer 2 is referring to in this comment, and thus the amendments they would like. We are happy to take editorial guidance here.

2. On page 27, the authors describe how the team decided to leave out detailed descriptions of withdrawal symptoms to avoid setting negative expectations, and instead referring patients to their GP for further guidance. However, on page 26 concerns were raised about the accessibility of GP care and how this could be a hindrance to proceed with discontinuation. The authors might reflect on this in

the discussion, and possibly explore alternative options (are there others who might provide this care and who are more easily contacted, preferably the same day?).

Thank you for highlighting this. We have added a comment into our results section where we discuss how access to GP appointments might be a barrier to patients receiving information about withdrawal symptoms.

Page 18 line 5-8:

While this information is provided to GPs as part of our health professional intervention package, it must be acknowledged that there are limitations around access to GP appointments which may act as a barrier to getting information about withdrawal symptoms.

3. The manuscript is generally written in a clear and fluent style, however at some points appears slightly colloquial. Some sentences could be improved to a more academic level.

Thank you for taking the time to review this manuscript. Throughout our revision, we have been mindful of the tone of voice and made sure the language is formal and academic.

Reviewer: 3

Reviewer Name

Dr Tom Kingstone

Institution and Country

School of Primary, Community and Social Care, Keele University, UK

Please state any competing interests or state 'None declared':
None declared.

Please leave your comments for the authors below Dear authors,

A fantastic body of work and a very commendable effort to fit a lot of information and process into this manuscript.

We are grateful for the positive comments from Reviewer 3.

I read with much interest and would like to suggest the following amendments:

Abstract: The statement of objectives in the abstract does not match those stated at the end of the intro/background section or the title of the paper. Please refer to optimisation for consistency.

Thank you for highlighting this. We have amended the abstract so that it is more consistent with our aims in the introduction.

Page 2 line 3-5:

We aimed to develop a digital intervention to support antidepressant discontinuation in UK primary care that is scalable, accessible, safe and feasible. In this paper we describe the development using a theory- evidence- and person-based approach.

Methods: description of the 'think aloud' method seems a little light in detail. For example, how much of the ADvisor intervention was covered in each interview? Were participants given a laptop or did they use their own PCs (important for familiarity, and recruitment)?

Thank you for this suggestion. We have added further details about the think aloud interview methods, including how much of the intervention was covered and that patients used a university laptop.

Page 8 line 1-14:

“Eligible participants met with a researcher (HB, SW or TK) either in their own home or at their primary care practice to take part in a think-aloud interview. Interviews invited participants to engage with the prototype intervention using a study laptop and say what they were thinking, aloud in real time. The interviewer prompted participants when necessary (for example asking patients ‘How do you feel about the information on this page?’). Interviews ranged from 38 to 93 minutes in length and were audio recorded, and transcribed verbatim. The interview ended when patients concluded they had looked at all the information they would like to see or if the interview length was approaching 90 minutes. The amount of intervention content the patient saw therefore depended on their own preferences and the time they took to look at the information. The interview schedule can be found in Appendix C. There were three primary iterations of interviews based on three key modified prototype interventions. Patients at the start of the study therefore saw different versions of the intervention to those who were recruited later rounds. This allowed the changes made as a result of patient feedback to continue to be tested.”

Recruitment: Do the authors mean >10 or ≥ 10 (suggesting moderate level of depression and/or depression not under control). Some details are missing in the sample characteristics: age, education level, level of internet use (if asked)?

We have made clear that patients were excluded if their PHQ-9 score was greater than or equal to 10. We have added age to our sample characteristics table (Table 3). However we did not collect data on education level or level of internet use.

Page 7 line 26-27:

Exclusion criteria: PHQ-9 scores greater than or equal to 10 (suggesting persisting symptoms of depression)...

Analysis: I was a little confused with the description of the two processes of analysis in the qualitative study. How did they overlap and/or inform one another? In terms of saturation, the definition seems to apply more to the rapid coding approach than to the thematic approach.

Thank you for highlighting this. We have provided further clarity about how the two approaches were used and how we decided when saturation was reached for both analyses.

Page 8 line 20-33:

“Transcribed interviews were analysed using two primary analytic methods. The first analytic method was a more rapid coding than thematic analysis, which involves using coding tables designed for the PBA, where positive and negative comments were tabulated. Core problematic issues likely to affect participant engagement or intervention effectiveness identified using this coding method were brought to the broader group, and amendments to the intervention agreed. Alongside this method, a more in-depth thematic analysis [26,27] was developed to capture patient views of the intervention and ideas

about how they might engage with it, beyond comments on what might be amended. For this latter analysis, HB independently coded the transcripts and discussed a preliminary coding frame with a second researcher (AG). Theme labelling and interpretation were discussed and agreed by the team. The thematic analysis is presented here. Therefore while the initial analysis informed what changes were necessary, the thematic analysis explored what patients thought about the intervention in greater depth. These analyses were related in that some things that were identified in our initial analysis informed the development of themes.”

Findings: I would have found it helpful to have another flowchart to present the key changes made to the ADvisor intervention across the interviews; to describe the different iterations and, most importantly, the process of optimisation (unless this is reported elsewhere?).

Reviewer two has suggested to keep the methods and results more concise. Therefore, instead of creating a flow chart detailing the interactions, we have provided a brief overview of the three rounds of interviews in the results section.

Page 13 line 17-25:

“There were three rounds of iterations of the intervention during the think-aloud interviews. Patients in round one were shown the first prototype. Changes made to the version in round two included making the tone less formal, revising the introduction navigation and the wording to be more gentle. The ‘my notes’ section was also reorganized to be clearer and buttons to exit the intervention at the end of each module were removed to try to keep the patients on site for longer. In the version shown in round three some changes included further revision of the tone, some of the information was presented in a more aesthetically pleasing way and some links within the intervention to other modules were removed as these were confusing for patients.”

VERSION 2 – REVIEW

REVIEWER	Marloes Huijbers Radboud University Medical Center, Nijmegen, the Netherlands
REVIEW RETURNED	05-Dec-2019

GENERAL COMMENTS	I appreciate the revisions of the authors in this manuscript. In particular, I think that revising the results section on the theory-part has made the manuscript more easy to read. As a reader unfamiliar with intervention planning theory, it still feels like quite a lot of theoretical models and it is not always clear to me how they translate to the actual content (which seems valid in itself based on the literature), or how these models add to the (validity of) the content (particularly Appendix B). However, as said I'm not an expert in intervention planning so I will refrain from further commenting on this topic. As an elaboration on the reviewer checklist: unless I have overlooked it in the manuscript: please include information about patient consent.
---

REVIEWER	Tom Kingstone School of Primary, Community and Social Care Keele University, UK
REVIEW RETURNED	22-Nov-2019

GENERAL COMMENTS	Dear authors Thank you for responding to my original comments and those of my fellow reviewers. I am satisfied with the changes that have been made. With kind regards
---

VERSION 2 – AUTHOR RESPONSE

#Reviewer: 2

1) “I appreciate the revisions of the authors in this manuscript. In particular, I think that revising the results section on the theory-part has made the manuscript more easy to read”.

Response: We are pleased this reviewer feels our amendments have improved the manuscript.

2) “As a reader unfamiliar with intervention planning theory, it still feels like quite a lot of theoretical models and it is not always clear to me how they translate to the actual content (which seems valid in itself based on the literature), or how these models add to the (validity of) the content (particularly Appendix B). However, as said I'm not an expert in intervention planning so I will refrain from further commenting on this topic.”

Response: We appreciate this point. We have now added text to provide an example of how the theory was applied on page 11, paragraph 4. See below:

“Fundamentally, SCT [32] underlies the approach taken in the intervention to facilitate behaviour change. We ensured content aligned with the principles of SCT on how best to increase patients confidence that they will be able to safely stop antidepressants (e.g. drawing on persuasion, modeling and supporting performance exposure). We also focused on modifying outcome expectations e.g. increase positive expectation that the recommended strategies are likely to support effective discontinuation.”

3) “As an elaboration on the reviewer checklist: unless I have overlooked it in the manuscript: please include information about patient consent.”

Response: We thank the reviewer for noting this. We have now added that written consent was obtained for the qualitative aspects on page 6, paragraph 2 and page 8, paragraph 2.

#Reviewer: 3

“Dear authors

Thank you for responding to my original comments and those of my fellow reviewers.

I am satisfied with the changes that have been made.

With kind regards”

Response: We are pleased that Reviewer 2 was happy with the changes we made.